# High Concordance of Genomic Profiles between Primary and Metastatic Colorectal Cancer

**DOI:** 10.3390/ijms22115561

**Published:** 2021-05-24

**Authors:** Seung Eun Lee, Ha Young Park, Dae-Yong Hwang, Hye Seung Han

**Affiliations:** 1Department of Pathology, School of Medicine, Konkuk University, Seoul 05030, Korea; 20150063@kuh.ac.kr; 2Department of Pathology, Busan Paik Hospital, College of Medicine, Inje University, Busan 47392, Korea; 3Department of Surgery, School of Medicine, Konkuk University, Seoul 05030, Korea; hwangcrc@kuh.ac.kr

**Keywords:** metastatic colorectal cancer, genomic profiles, high concordance between primary and metastatic colorectal cancer

## Abstract

The comparison of the genetic profiles between primary and metastatic colorectal cancer (CRC) is needed to enable the discovery of useful therapeutic targets against metastatic CRCs. We performed the targeted next generation sequencing assay of 170 cancer-associated genes for 142 metastatic CRCs, including 95 pairs of primary and metastatic CRCs, to reveal their genomic characteristics and to assess the genetic heterogeneity. The most frequently mutated gene in primary and metastatic CRCs was *APC* (71% vs. 65%), *TP53* (54% vs. 57%), *KRAS* (45% vs. 44%), *PIK3CA* (16% vs. 19%), *SMAD4* (15% vs. 14%) and *FBXW7* (11% vs. 11%). The concordance in the top six frequently mutated genes was 85%, on average. The overall mutation frequencies were consistent with two sets of public data (TCGA and MSKCC). To the author’s knowledge, this is the first study to compare the genetic profiles of our cohort with that of the metastatic CRCs from MSKCC. Comparative sequencing analysis between primary and metastatic CRCs revealed a high degree of genetic concordance in the current clinically actionable genes. Therefore, the genetic investigation of archived primary tumor samples with the challenges of obtaining an adequate sample from metastatic sites appears to be sufficient for the application of cancer precision medicine in the metastatic setting.

## 1. Introduction

Colorectal cancer (CRC) is one of the well-established tumor types to be considered as a genetic disease in which the multiple and sequential accumulation of genetic alterations underlies the development and progression to carcinoma and metastasis. The inactivation of *APC* mutations, activation of *KRAS* mutations, and the diverse mutations in *TP53*, *PIK3CA*, and *SMAD4*, TGF-β pathway genes, drive the development and evolution of a malignant CRC [1].

A comprehensive investigation of the genomic landscape of the early stages of CRC, was reported by The Cancer Genome Atlas (TCGA) Network Network [2]. More recently, metastatic CRCs were also analyzed by using MSK-IMPACT, a capture-based next generation sequencing (NGS) platform [3]. Several studies have performed the analysis of the comparative genetic sequencing of paired primary and metastatic CRC [4,5,6,7]. The majority have shown a high degree of concordance in the genetic profile between primary and metastatic CRC [4,5,6]. Especially, early recurrent genetic alterations such as *APC*, *KRAS*, *NRAS*, and *BRAF*, involving colorectal carcinogenesis, were highly concordant in matched pairs of primary and metastatic CRC [5,6].

Genetic intratumor heterogeneity within tumors often occurs as a result of the progressive accumulation of genetic alterations during the spatial and temporal evolution of the tumor. More recently, several studies have reported that advanced CRCs harbor extensive intratumor heterogeneity, shaped by neutral evolution during tumor evolution [8,9]. Furthermore, Saito et al. demonstrated that the evolutionary principle shaping genetic intratumor heterogeneity shifts from Darwinian to neutral evolution during CRC progression [9].

The comparison of the genetic profile between primary and metastatic CRCs is needed to enable the discovery of useful therapeutic targets against metastatic CRCs. In this study, we performed the targeted next generation sequencing (NGS) assay of 170 cancer-associated genes for 142 metastatic CRCs, including 95 pairs of primary and metastatic colorectal samples, to reveal their genomic characteristics and to assess the genetic heterogeneity between primary and metastatic CRCs.

## 2. Results

### 2.1. Clinicopathologic Characteristics of Patients

A total of 146 patients were analyzed in this study. From 95 patients, pairs of primary and metastatic colon cancer samples were analyzed to compare the genetic profiles. An additional 4 primary only and 47 metastatic only samples were also analyzed. The clinicopathologic characteristics of 142 metastatic tumors (including 95 paired with primary tumor and 47 singleton) are summarized in Table 1. The median age at diagnosis was 61 years (range, 34–89). The most common sites of metastasis were the liver (54.2%) and the lung (24.7%), followed by the abdominopelvic cavity (12.7%) and central nervous system and soft and bone metastases were rare. In total, 78 (54.9%) and 64 (45.1%) of 142 patients developed synchronous and metachronous metastasis, respectively. Only 4.9% of metastatic CRCs displayed an MSI-H genotype/phenotype.

### 2.2. Genomic Profiling of 95 Paired Samples

After the filtering steps to identify the clinically significant mutations described in the Method section, a total of 318 mutations of 51 genes, including 243 missense and nonsense single nucleotide variants (SNVs), 52 insertion and deletions (indels) and 23 splice site mutations were identified in 95 primary-tumor pairs. The full list of variants can be found in Appendix A. Variants occurred in more than 3 out of 95 pairs and are summarized in Figure 1. Of the 318 mutations, 81% (258/318) were found in both primary and metastatic CRCs, 12% (37/318) were found only in primary CRCs, and 7% (23/318) were found only in metastatic CRCs. Concordance was different according to the type of variant, 85% for SNVs, 73% for indels and 57% for splice site mutations. The concordance of the variants in the top six genes was 85% on average, the lowest in SMAD4 was 61% and the highest in *FBXW7* was 100%.

The average number of concordant variants per sample was 5.4 and discordant variants per sample was 0.6. The discordant variants were caused more frequently by primary specific variants (37/60, 62%), compared with the metastatic-specific variants (23/60, 38%). The portion of concordant mutations was high in *FBXW7*, *NRAS*, *PTEN*, and *BRCA2* (100% concordance) gene, whereas the portion of discordant mutations was high in *ERBB2*, *PIK3R1*, *TSC1*, and *VHL* (concordance ≤40%) genes. The mean concordance of variants in all 95 paired primary and metastatic CRCs was 87%. Concordance was significantly higher in the synchronous group than in the metachronous group (92% vs. 86%, *p* = 0.0124). The ratio of primary specific variants was significantly lower in the synchronous group than in the metachronous group (4% vs. 11%, *p* = 0.0016).

The most frequently mutated gene in primary CRCs was *APC* (71%, 67/95), followed by *TP53* (54%, 52/95), *KRAS* (45%, 43/95), *PIK3CA* (16%, 15/95), *SMAD4* (15%, 14/95) and *FBXW7* (11%, 10/95). In metastatic CRCs, the order was the same as the primary group, *APC* (65%, 62/95), followed by *TP53* (57%, 54/95), *KRAS* (44%, 42/95), *PIK3CA* (19%, 18/95), *SMAD4* (14%, 13/95) and *FBXW7* (11%, 10/95). The frequency of the top 6 gene mutations was almost the same in primary and metastatic CRCs (Figure 2A). The frequency of mutations was numerically higher in primary tumors than in metastatic tumors in *APC*, although, there was no overall statistical difference in the frequency of mutations between primary and metastatic tumors.

We also compared the variant allele frequency (VAF) of the frequently mutated top six genes between primary and metastatic CRCs. (Figure 2B and Table 2). Of the six genes, VAFs of *TP53* mutations in metastatic CRCs were higher than those of primary CRCs (0.24% ± 0.15% in primary CRCs vs. 0.33% ± 0.22% in metastatic CRCs), which was statistically significant by paired two sample t-test (*p* = 0.0024).

No discordant mutation in *FBXW7* was observed and most of the VAFs in metastatic CRCs were lower than those of primary CRCs (9/11), which is tumor cells harboring a *FBXW7* mutation might be subclonal in metastatic CRCs. No obvious differences in the VAFs of *APC*, *KRAS* and *PIK3CA* were detected between primary and metastatic CRCs.

Mutational counts per sample did not show significant differences according to the patterns of metastasis (synchronous vs. metachronous), location of metastasis (liver vs. others) and MSI status.

### 2.3. Genomic Profiling of All 142 Samples Including Primary/Metastatic Singletons

When analyzing all samples, including unpaired primary or metastatic only in each primary and metastatic group, the top six genes were the same in both groups (Figure 3). The variants in *APC* were mostly the truncating type, 72% nonsense SNVs, 26% indels and 2% splice site mutations in the metastatic group. For *TP53*, 66% were missense SNVs encoding p.R175H (16%), p.R248Q/W (8%) or p.R273H/S (5%) altered proteins and 34% were the truncating type including 22% nonsense, 8% indels and 4% splice site mutations in metastatic group. All of the variants of the two oncogenes, *KRAS* and *PIK3CA* occurred in mutational hotspots. The mutational frequencies of the genes, except for the top six genes, were less than 5% and the rank of the mutational frequencies showed no significant difference. The genes with variants exclusively in the metastatic group were *AKT1*, *CDH1*, *BAP1*, *DNMT3A*, *FOXL2*, and *MSH6*. These genes are probable tumor suppressor genes according to the Cancer Gene Census [10], except for *AKT1*.

The clinicopathological factors showed statistically significant differences by mutational status. (Table 3) MSI-H ratio was higher in the *SMAD4* mutated group. The frequency of *TP53* mutations in synchronous metastasis were higher than those of metachronous metastasis. Overall survival was not different according to mutational status of the top six genes. Recurrence occurred significantly earlier in patients with *FBXW7* mutant (137 days in *FBXW7* mutant vs. 294 days in wild type, *p* = 0.0221).

### 2.4. Comparing with Public Data: 99 Primary CRCs vs. TCGA and 142 Metastatic CRCs vs. MSKCC

We compared our data with public cancer datasets from cBioPortal. (https://www.cbioportal.org/datasets, accessed on 19 May 2021) ‘Colorectal Adenocarcinoma (TCGA, Firehose Legacy)’ (TCGA, *n =* 223) and ‘Metastatic Colorectal Cancer (MSKCC, Cancer Cell 2018)’ (MSKCC, *n =* 1134) for primary and metastatic control, respectively. The top 20 genes of each group are listed in Table 4. Of the top 20 genes in TCGA or MSKCC, *AMER1*, *SOX9*, *ARID1A* and *TCF7L2* were not included in our cancer panel. TCGA, the datasets with genetic profiles of primary CRCs and MSKCC, those of metastatic CRCs were compared with the results in this study (Figure 4).

When comparing two public datasets, TCGA and MSKCC, the genes with significantly lower mutational frequency in MSKCC than in TCGA were *AMER1* (9% vs. 4%, *p* = 0.0326) and *NRAS* (8% vs. 3%, *p* = 0.0463). The genes with significantly higher mutational frequency in MSKCC than in TCGA were *TP53* (53% vs. 72%, *p* < 0.0001), *PIK3CA* (13% vs. 19%, *p* = 0.0188), and *SOX9* (4% vs. 9%, *p* = 0.0088). The genes showing significant difference between two public datasets were *NRAS*, *TP53*, and *PIK3CA*, and these genes showed similar mutational frequencies between the primary and metastatic CRCs of the present study.

When comparing the primary group of the present study with TCGA, the genes showing a difference in variant rate ≥5% were SMAD4 (15% vs. 10%, *p* = 0.1995), *FBXW7* (10% vs. 15%, *p* = 0.2225), and *BRAF* (2% vs. 9%, *p* = 0.0022). Between the metastatic group of the present study and MSKCC, *APC* (63% vs. 73%, *p* = 0.0129), *TP53* (51% vs. 72%, *p* < 0.0001), *PIK3CA* (14% vs. 19%, *p* = 0.1209), and *BRAF* (3% vs. 11%, *p* < 0.0001) showed differences of more than 5%.

*APC* mutations were mostly the truncated type in MSKCC (1268/1217 mutations, 99.8%) as in our results of the present data. In terms of types of variants, indel ratio was significantly higher in MSKCC than our metastatic group (41% vs. 26%, *p* = 0.0004). Considering *TP53* mutations, ratio of missense SNVs were 66% in both the metastatic group of the present study and MSKCC, and nonsense SNV ratio was higher in the metastatic group of the present study than in MSKCC (22% vs. 14%). However, indel ratio was significantly lower in the metastatic group of the present study than in MSKCC (8% vs. 15%, *p* = 0.0488). *PIK3CA* mutations occurred at p.H1047 were 36% in the metastatic group of the present study and 17% in MSKCC, at p.E545 were 32% vs. 27%. Ninety-five percent of *PIK3CA* variants were found in the amino acid positions of 1047, 542, 545 and 546 in the metastatic group of the present study, 32% of variants were found at other positions in MSKCC. *BRAF* V600E mutation was only found in two patients of the present study.

## 3. Discussion

We performed targeted NGS of 170 cancer-associated genes in 142 metastatic CRCs, including 95 pairs of primary and metastatic CRCs to define the mutational concordance of these genes in primary and metastatic CRCs. Furthermore, we compared our data with the public cancer datasets, TCGA (*n =* 223) [2] and MSKCC (*n =* 1134) [3]. The previous comparative sequencing studies between primary and metastatic CRCs were only compared with the data of TCGA [6,11]. TCGA analyzed only the primary CRCs, the majority of which were derived from the early stages of CRCs [2]. Meanwhile, our study cohort, which consists of all patients with metastatic CRCs, and the MSKCC CRC cohort, which also had more aggressive and advanced CRCs, were distinct from the TCGA cohort [3]. Genomic analysis in the MSKCC thus provided insights into more metastatic CRCs that were not evident in the TCGA CRCs cohort. Therefore, we compared genetic profiles of primary CRCs in our study with TCGA, and those of the metastatic CRCs with the MSKCC in our study, respectively. Our data is significant in comparing primary and metastatic CRCs in pairs, as well as comparing two large public data, TCGA and MSKCC, representative of primary and metastatic CRCs, under the same conditions. To the author’s knowledge, this is the first study to compare the genetic profile of our cohort with that of the metastatic CRCs from MSKCC dataset.

We identified that the frequency of recurrent mutations of *APC*, *KRAS*, *PIK3CA*, *FBXW7*, and *SMAD4* were consistent with previous reports on metastatic CRCs [3,12,13]. Overall concordance of the clinically significant mutations between primary and metastatic CRCs was 81%. This concordance rose to 85% for the six most recurrent mutations occurring in CRCs. The six most recurrent mutations were known as the CRC driver genes. These results are consistent with those of previous studies on comparative sequencing [3,5,14]. In MSKCC, a high level of genomic concordance was also identified between the primary and metastatic CRCs [3]. Therefore, a low degree of genetic heterogeneity between primary and metastatic CRCs with respect to driver mutations of CRCs. This supported that the main driver genetic alterations involving colorectal carcinogenesis was maintained during the evolution of tumor metastasis. Among the driver genes, the concordance was high in *KRAS* (98%), *NRAS* (100%), *APC* (90%), which are early or universal mutations in CRCs. On the other hand, the concordance was relatively low in *PIK3CA* (70%), *SMAD4* (61%), which are known as later mutations. This observation was consistent with prior studies [11] and could be explained through heterogeneous clonal evolution [15].

Especially, our result showed that with *KRAS* (98%), *NRAS* (100%), there was very high concordance between primary and metastatic CRCs. A meta-analysis of all published studies between 1991 and 2018 reporting on biomarker concordance between primary and metastatic CRCs, a very high median biomarker concordance for *KRAS* (93%), *NRAS* (100%), *BRAF* (99.4%), *PIK3CA* (93%) was reported, whereas meta-analytic pooled discordance was 8% for *KRAS*, 8% for *BRAF*, and 7% for *PIK3CA* in 61 studies, including 3565 patient samples [16]. *KRAS* and *NRAS* are OncoKB level-1 resistance biomarkers for anti-EGFR (epidermal growth factor receptor) antibodies (cetuximab and panitumumab), mutational testing of these genes has now been incorporated into the National Comprehensive Cancer Network (NCCN) guidelines for the treatment of patients with metastatic CRCs [17]. These results are also consistent with the recommendation that molecular analysis of the primary tumor is representative of the genetic characteristic of the metastatic tumor [17]. Therefore, the genetic investigation of archived primary tumor samples with challenges in getting adequate samples from metastatic sites appears to be sufficient for the application of cancer precision medicine in the metastatic setting.

Concordance between primary and metastatic CRCs was significantly higher in the synchronous group than in the metachronous group. Synchronous metastatic tumors were mainly treatment-naïve while the majority of patients with metachronous metastatic tumors have received systemic treatment such as chemotherapy for the primary tumor, which may influence the mutational concordance between the primary and the subsequent tumor.

The genes with significantly lower mutational frequency in the metastatic group of the present study than in the MSKCC were *TP53*, and *APC*, whereas genes showing significant difference of mutational frequency between the primary group of the present study and TCGA were not identified. Considering the distribution of the types of variants, the MSKCC group showed a significantly higher proportion of indels among the total variants than those of the present study and TCGA. (35% for MSKCC, 14% for present study and TCGA, *p* < 0.0001). *TP53* and *APC* are the most representative tumor suppressor genes which commonly suffer loss of their function by deletion mutations. In the MSKCC group, due to a more efficient indel calling algorithm, more indels may have been detected than in the present study and made a difference in the mutational frequencies in the two genes commonly altered by deletion mutation.

*TP53* alterations in the MSKCC group, the genetic profiles of metastatic CRCs, were the only genomic alteration significantly enriched in metastatic CRCs, which shows *TP53* alterations are selectively enriched in metastatic CRCs [3]. In our study, VAFs of *TP53* mutations in metastatic CRCs were significantly higher than those of the primary CRCs, although there was no overall statistical difference in the frequency of mutations between paired primary and metastatic CRCs. This result could be interpreted that the clones with the *TP53* mutation might be expanded through sustained tumor growth and metastasis or an additional genetic hit, resulting in loss of heterozygosity.

*SMAD4*, a downstream regulator in the TGF-β signaling pathway in CRC, has been highlighted. In particular, inactivation of *SMAD4* has been associated with late stage or metastatic CRCs [18]. Recent work has also highlighted that *SMAD4* downregulation may occur in up to 60% of patients with metastatic CRC, which is significantly higher than the incidence of *SMAD4* mutations [19]. However, there was no statistical difference in the frequency of *SMAD4* mutations between primary and metastatic CRCs in our study.

*FBXW7* is a tumor suppressor gene and the frequency of mutation in CRCs has been reported at 6–10% [20,21,22]. It was recently reported that *FBXW7* mutations had significantly worse survival in metastatic CRCs [23]. In our study, recurrence occurred significantly early in patients with the *FBXW*7 mutant than in patients with wild type, although the overall survival (OS) was not different according to *FBXW*7 mutational status.

There was a lower prevalence of *BRAF* genes in the both primary (2%) and metastatic (3%) CRCs of the present study compared with TCGA (9%) and MSKCC (11%) data. According to a previous review, the prevalence of *BRAF*-mutated CRC is lower in Eastern Asian countries (0.7–11.4%) than in Western countries (3.7–20.6%) [24].

Only 4.9% of metastatic CRCs displayed an MSI-H genotype/phenotype, a frequency that is lower than that reported for primary CRCs but similar to the MSK cohort (4%) [3]. These findings could be explained by the lower tendency of MSI-H genotype/phenotype tumors to metastasize [25].

The limitation of this study is that we could not analyze the difference of frequency and concordant rate of genes between primary and metastatic CRCs according to MSI-status due to the too small size of the MSI-H genotype/phenotype tumors. Therefore, future studies, with a large cohort of MSI-H of metastatic CRCs, are needed to further investigate the difference of frequency and concordance in MSS and MSI-H group. A second limitation is that our study only examined the changes in DNA. The areas that are not well explained about the association with clinicopathological factors may be explained by other aspects of genetic variation, such as methylation and microenvironment.

## 4. Materials and Methods

### 4.1. Patients

A total of 142 patients with CRC and distant metastasis were enrolled in this study. Ninety-five cases were available for both primary and metastatic tumor tissues. Additional 47 metastatic only samples were included for mutational profiling of metastatic tumors and 4 primary only samples were used as a control for metastatic tumors. H&E stained slides were reviewed by pathologists and samples with tumor cellularity ≥50% were included to detect confident variants. Clinicopathological information, including age, sex, smoking history, and stage of cancer, was obtained retrospectively by reviewing medical records. Synchronous metastasis was defined as a metastatic disease at the time or within 6 months of the original diagnosis of CRC. Metachronous metastasis was defined as the absence of metastatic disease at the time of the initial diagnosis with metastatic disease developing later than 6 months from the original diagnosis. The study protocol was approved by the institutional review board of Konkuk University Medical Center (KUH1210052), and written informed consents were obtained from all patients.

### 4.2. Targeted Sequencing & Identification of Clinically Significant Mutations

Custom panel including 170 cancer genes (Appendix A) was used for mutational profiling. Sample preparation, sequencing procedures, and variant calling pipeline were almost same as the one described in our previous publication [26]. After variant calling, variant annotations were done using VEP [27] to annotate genes, variant type, gnomAD [28] minor allele frequency (MAF), and ClinVar [29] significance. Annotation with OncoKB [30] variant annotator (https://github.com/oncokb/oncokb-annotator, accessed on 1 September 2020) was also performed to check oncogenicities of variants.

To identify confident and clinically significant variants, we applied filtering criteria as followings: (1) protein altering variants including missense and nonsense SNVs, splice site mutations and indels, (2) rare variants with MAF of gnomAD total and east Asian ≤0.1%, (3) probable pathogenic mutations with ClinVar significance in one of ‘Pathogenic’, ‘Likely pathogenic’, or ‘Drug response’, or oncoKB annotation with one of ‘Predicted oncogenic’, ‘Likely oncogenic’, or ‘Oncogenic’.

### 4.3. Comparing with Public Data

To compare mutational frequencies of primary and metastatic tumors with public data, we downloaded files from cBioPortal. (https://www.cbioportal.org/datasets, accessed on 19 May 2021) Annotated variants of dataset ‘Colorectal Adenocarcinoma (TCGA, Firehose Legacy)’ (*n =* 223) were compared with primary tumor group and ‘Metastatic Colorectal Cancer (MSKCC, Cancer Cell 2018)’ (*n =* 1134) were compared with metastatic tumor group. To compare public data with our results under the same conditions, ClinVar and OncoKB annotation were performed for the two datasets and filtering steps by the criteria described above were applied.

### 4.4. Molecular Findings for Microsatellite Instability (MSI)

We performed an MSI analysis on paraffin-embedded tissues to evaluate MSI status. The MSI status of the tumor samples was determined by using the five-marker Bethesda panel (BAT25, BAT26, D5S346, D2S123 and D17S250) [31]. Polymerase chain reaction (PCR) products were run on a Qsep 100 DNA fragment analyzer (Bioptic Inc., Taiwan, China) and analyzed using Qsep 100 viewer (Bioptic Inc., Taiwan, China). Microsatellite instability was defined by the presence of different sized alleles in tumor DNA compared with the matched normal DNA sample. We classified the results into microsatellite instability-high (MSI-H), microsatellite instability–low (MSI-L) and microsatellite stable (MSS) in tumors according to Bethesda guidelines [32].

### 4.5. Statistical Analysis

The association between mutational status and clinicopathological features was analyzed with the χ2 test or Fisher’s exact test. Paired two sample t-test was done by comparing the VAFs of 95 pairs of primary and metastatic tumors. Two sample proportion test was used in comparing mutational frequencies between groups. The overall survival (OS) was the primary endpoint for this study and was calculated from the date of surgery until the date of death. The Kaplan−Meier method was used to estimate the OS. *p*-value of less than 0.05 was considered to indicate a statistically significant difference. All analyses were carried out using Rex software version 3.0.3 (RexSoft Inc., Seoul, Korea).

## 5. Conclusions

In conclusion, our data is significant in comparing with primary and metastatic CRCs in pairs, as well as comparing two large public data, TCGA and MSKCC, representative of primary and metastatic CRCs, under the same conditions. To the author’s knowledge, this is the first study to compare the genetic profiles of our cohort with that of the metastatic CRCs from MSKCC dataset. Comparative sequencing analysis between primary and metastatic CRCs revealed a high degree of genetic concordance in the main driver genes, especially, the current clinically actionable genes. Therefore, the genetic investigation of archived primary tumor samples with the challenges of obtaining adequate samples from metastatic sites appears to be sufficient for the application of cancer precision medicine in the metastatic setting.

## Figures and Tables

**Figure 1 ijms-22-05561-f001:**
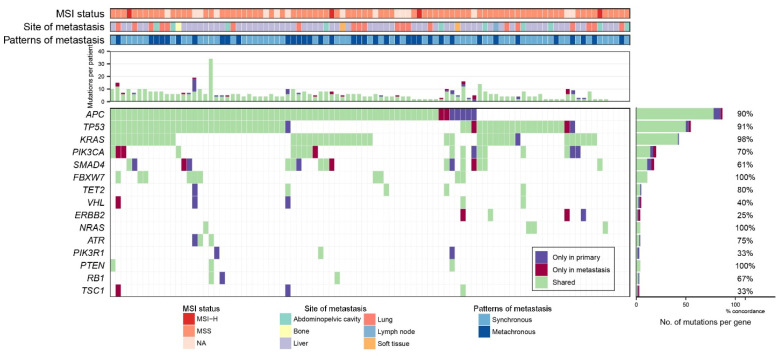
Overview of mutational profile of 95 pairs of primary-metastatic colorectal cancers. Each column represents a patient. The top 4 panels show MSI status, site of metastasis, patterns of metastasis and number of variant counts per patients. The bottom panel show mutational status, shared, only in primary and only in metastasis. When there are 2 or more mutation type presents, (i.e., ‘shared’ and ‘primary only’) ‘shared’ is preferentially used. The right panel represents the number of patients with variants as bar graph and concordance of the variants as percentage of corresponding genes.

**Figure 2 ijms-22-05561-f002:**
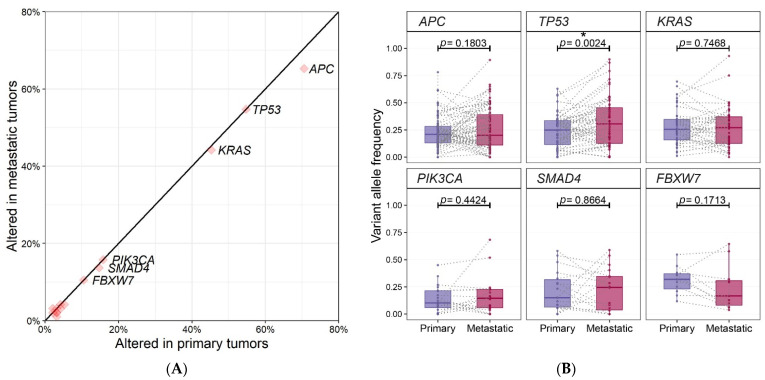
(**A**) Comparison of mutational frequencies between 95 pairs of primary and metastatic tumors. It is the proportion of patients with mutations in the gene. Top six genes are annotated. (**B**) Comparison of variant allele frequencies of top six genes between 95 pairs of primary and metastatic tumors. Boxplots show the distribution of VAFs in each group and dots connected by dashed-lines represent same patients. Statistical significance is based on paired two sample *t*-test and *p* < 0.05 is marked with asterisk (*).

**Figure 3 ijms-22-05561-f003:**
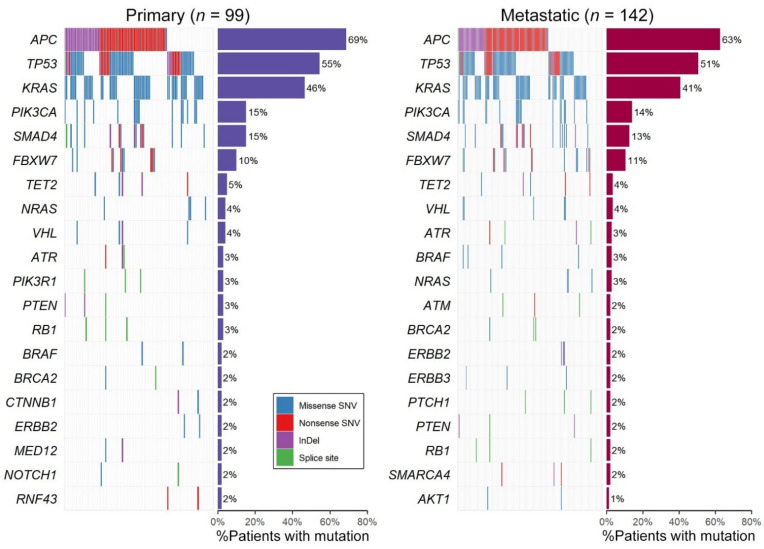
Landscape of mutations in 99 primary and 142 metastatic groups (including 95 pairs of primary and metastatic tumors). Each column represents a single patient. The percentage of mutations across each group is represented by vertical histograms.

**Figure 4 ijms-22-05561-f004:**
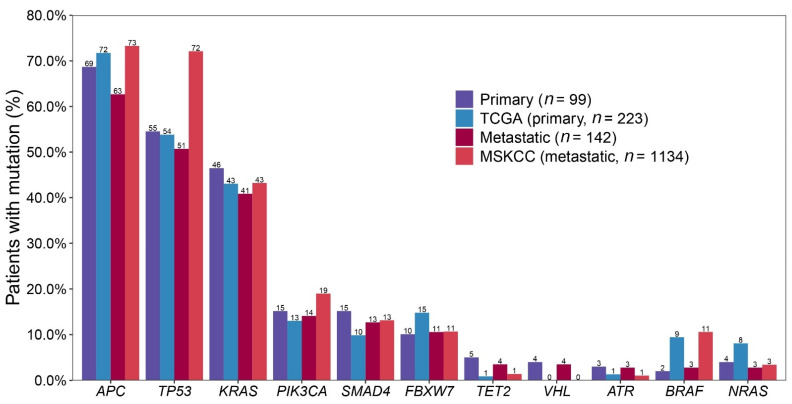
Most commonly mutated gene frequencies of present study compared with TCGA dataset as control for 99 primary group, and MSKCC dataset as control for 142 metastatic group.

**Table 1 ijms-22-05561-t001:** Clinicopathologic characteristics of 142 metastatic tumors.

Variable	Category	Total	
		(*n =* 142)	
Age			
	Mean ± SD	60.51 ± 11.17	
	Median	61	
	Range (min–max)	34–89	
Gender			
	Male	95	66.90%
	Female	47	33.10%
Site of metastasis			
	Liver	77	54.23%
	Lung	35	24.65%
	Abdominopelvic cavity	18	12.68%
	Soft tissue	6	4.23%
	Brain	2	1.41%
	Lymph node	2	1.41%
	Bone	1	0.70%
	Skin	1	0.70%
Location of primary cancer			
	Ascending colon	16	11.30%
	Hepatic flexure	6	4.20%
	Transverse colon	5	3.50%
	Splenic flexure	2	1.40%
	Descending colon	6	4.20%
	Rectosigmoid	107	75.40%
Patterns of metastasis			
	Synchronous	78	54.90%
	Metachronous	64	45.10%
T stage			
	1	3	2.11%
	2	7	4.93%
	3	101	71.13%
	4	26	18.31%
	NA	5	3.52%
N stage			
	0	39	27.46%
	1	48	33.80%
	2	48	33.80%
	NA	7	4.93%
Grade			
	1	2	1.41%
	2	132	92.96%
	3	4	2.82%
	NA	4	2.82%
Vascular invasion			
	Absent	117	82.39%
	Present	20	14.08%
	NA	5	3.52%
Lymphatic invasion			
	Absent	72	50.70%
	Present	65	45.77%
	NA	5	3.52%
Perineural invasion			
	Absent	97	68.31%
	Present	40	28.17%
	NA	5	3.52%
MSI status			
	MSI-H	7	4.93%
	MSS	107	75.35%
	NA	21	14.79%

**Table 2 ijms-22-05561-t002:** Variant allele frequency (VAF) of frequently mutated top six genes between primary and metastatic CRCs.

Genes	No. of variants	VAF of Primary Tumors (Mean ± SD)	VAF of Metastatic Tumors (Mean ± SD)	*p*-Value
*APC*	87	0.23 ± 0.14	0.26 ± 0.19	0.1803
*TP53*	55	0.24 ± 0.15	0.33 ± 0.22	0.0024
*KRAS*	43	0.28 ± 0.16	0.27 ± 0.19	0.7468
*PIK3CA*	20	0.14 ± 0.12	0.17 ± 0.17	0.4424
*SMAD4*	18	0.21 ± 0.19	0.22 ± 0.20	0.8664
*FBXW7*	11	0.31 ± 0.12	0.24 ± 0.21	0.1713

**Table 3 ijms-22-05561-t003:** Association between mutational status and clinicopathological features.

Mutational Status	Total	MSI Status		*p* Value	Total	Location of Primary Cancer		*p* Value	Total	Patterns of Metastasis		*p* Value
	Patients	MSI-H	MSS		Patients	Liver	Others		Patients	Synchronous	Metachronous	
	(*n =* 118)	(*n =* 7)	(*n =* 111)		(*n =* 142)	(*n =* 77)	(*n =* 65)		(*n =* 142)	(*n =* 78)	(*n =* 64)	
*APC*												
Wild type (%)	47 (39.83)	3 (42.86)	44 (39.64)	>0.99	53 (37.32)	24 (31.17)	29 (44.62)	0.14	53 (37.32)	32 (41.03)	21 (32.81)	0.41
Mutated (%)	71 (60.17)	4 (57.14)	67 (60.36)		89 (62.68)	53 (68.83)	36 (55.38)		89 (62.68)	46 (58.97)	43 (67.19)	
*TP53*												
Wild type (%)	58 (49.15)	5 (71.43)	53 (47.75)	0.27	70 (49.3)	33 (42.86)	37 (56.92)	0.13	70 (49.3)	32 (41.03)	38 (59.38)	0.04
Mutated (%)	60 (50.85)	2 (28.57)	58 (52.25)		72 (50.7)	44 (57.14)	28 (43.08)		72 (50.7)	46 (58.97)	26 (40.62)	
*KRAS*												
Wild type (%)	69 (58.47)	3 (42.86)	66 (59.46)	0.45	84 (59.15)	49 (63.64)	35 (53.85)	0.31	84 (59.15)	52 (66.67)	32 (50)	0.07
Mutated (%)	49 (41.53)	4 (57.14)	45 (40.54)		58 (40.85)	28 (36.36)	30 (46.15)		58 (40.85)	26 (33.33)	32 (50)	
*PIK3CA*												
Wild type (%)	99 (83.9)	6 (85.71)	93 (83.78)	>0.99	122 (85.92)	64 (83.12)	58 (89.23)	0.42	122 (85.92)	68 (87.18)	54 (84.38)	0.81
Mutated (%)	19 (16.1)	1 (14.29)	18 (16.22)		20 (14.08)	13 (16.88)	7 (10.77)		20 (14.08)	10 (12.82)	10 (15.62)	
*SMAD4*												
Wild type (%)	102 (86.44)	3 (42.86)	99 (89.19)	0.01	124 (87.32)	66 (85.71)	58 (89.23)	0.71	124 (87.32)	67 (85.9)	57 (89.06)	0.76
Mutated (%)	16 (13.56)	4 (57.14)	12 (10.81)		18 (12.68)	11 (14.29)	7 (10.77)		18 (12.68)	11 (14.1)	7 (10.94)	
*FBXW7*												
Wild type (%)	106 (89.83)	7 (100)	99 (89.19)	>0.99	127 (89.44)	66 (85.71)	61 (93.85)	0.19	127 (89.44)	68 (87.18)	59 (92.19)	0.49
Mutated (%)	12 (10.17)	0 (0)	12 (10.81)		15 (10.56)	11 (14.29)	4 (6.15)		15 (10.56)	10 (12.82)	5 (7.81)	

**Table 4 ijms-22-05561-t004:** Top 20 frequently mutated genes in 4 groups.

Primary (*n =* 99)				Metastatic (*n =* 142)				TCGA (*n =* 223)				MSKCC (*n =* 1134)			
Order	Gene	# Mutated Patients	% Mutation	Order	Gene	# Mutated Patients	% Mutation	Order	Gene	# Mutated Patients	% Mutation	Order	Gene	# Mutated Patients	% Mutation
1	*APC*	68	69%	1	*APC*	89	63%	1	*APC*	160	72%	1	*APC*	831	73%
2	*TP53*	54	55%	2	*TP53*	72	51%	2	*TP53*	120	54%	2	*TP53*	818	72%
3	*KRAS*	46	46%	3	*KRAS*	58	41%	3	*KRAS*	96	43%	3	*KRAS*	490	43%
4	*PIK3CA*	15	15%	4	*PIK3CA*	20	14%	4	*FBXW7*	33	15%	4	*PIK3CA*	215	19%
5	*SMAD4*	15	15%	5	*SMAD4*	18	13%	5	*PIK3CA*	29	13%	5	*SMAD4*	149	13%
6	*FBXW7*	10	10%	6	*FBXW7*	15	11%	6	*SMAD4*	22	10%	6	*FBXW7*	121	11%
7	*TET2*	5	5%	7	*TET2*	5	4%	7	*BRAF*	21	9%	7	*BRAF*	120	11%
8	*NRAS*	4	4%	8	*VHL*	5	4%	8	*AMER1*	19	9%	8	*SOX9*	99	9%
9	*VHL*	4	4%	9	*ATR*	4	3%	9	*NRAS*	18	8%	9	*ARID1A*	83	7%
10	*ATR*	3	3%	10	*BRAF*	4	3%	10	*ARID1A*	12	5%	10	*TCF7L2*	77	7%
11	*PIK3R1*	3	3%	11	*NRAS*	4	3%	11	*ATM*	10	4%	11	*PTEN*	68	6%
12	*PTEN*	3	3%	12	*ATM*	3	2%	12	*ERBB3*	10	4%	12	*RNF43*	67	6%
13	*RB1*	3	3%	13	*BRCA2*	3	2%	13	*SOX9*	10	4%	13	*AMER1*	49	4%
14	*BRAF*	2	2%	14	*ERBB2*	3	2%	14	*SMAD2*	9	4%	14	*ATM*	45	4%
15	*BRCA2*	2	2%	15	*ERBB3*	3	2%	15	*PIK3R1*	8	4%	15	*PIK3R1*	40	4%
16	*CTNNB1*	2	2%	16	*PTCH1*	3	2%	16	*CREBBP*	7	3%	16	*NRAS*	39	3%
17	*ERBB2*	2	2%	17	*PTEN*	3	2%	17	*FBN1*	7	3%	17	*B2M*	37	3%
18	*MED12*	2	2%	18	*RB1*	3	2%	18	*PTEN*	7	3%	18	*BRCA2*	35	3%
19	*NOTCH1*	2	2%	19	*SMARCA4*	3	2%	19	*BCOR*	6	3%	19	*SMAD3*	32	3%
20	*RNF43*	2	2%	20	*AKT1*	2	1%	20	*ELF3*	6	3%	20	*TGFBR2*	30	3%

The top six genes were common in all groups, the order showed differences in TCGA.

## Data Availability

Not applicable.

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
