# Peer review of "High Concordance of Genomic Profiles between Primary and Metastatic Colorectal Cancer"

_ijms, 2021, doi:10.3390/ijms22115561_

Round 1

Reviewer 1 Report

Comments

The authors analyzed the genomic profiles of 142 metastatic colorectal cancer patients including 95 pairs of primary and metastatic lesions. The majority of this paper has been suitably revised.

I have only minor points that I would like to see addressed for further improvement of this manuscript.

Minor Comments:

I would like to recommend including the explanations or citations of figures1, 2, and 3 in the main text.

Author Response

Reviewer(s)' Comments to Author: 

The authors analyzed the genomic profiles of 142 metastatic colorectal cancer patients including 95 pairs of primary and metastatic lesions. The majority of this paper has been suitably revised.

I have only minor points that I would like to see addressed for further improvement of this manuscript.

Minor Comments:

I would like to recommend including the explanations or citations of figures1, 2, and 3 in the main text.

Response: As advised, we already have added the explanations and citations of figure 1, 2, and 3 in the main text. We highlighted changes made in the revised manuscript.

Reviewer 2 Report

I appreciate the authors' efforts to improve the manuscript and underlying the novelty of the presented results. Please correct the supplementary materials formating, the tables should also follow the IJMS rules.

Author Response

Reviewer(s)' Comments to Author: 

I appreciate the authors' efforts to improve the manuscript and underlying the novelty of the presented results. Please correct the supplementary materials formating, the tables should also follow the IJMS rules.

Response: We agree with the reviewer’s opinion. We revised the supplementary Tables and added Table A1, and Table A2. We highlighted changes made in the revised manuscript.

This manuscript is a resubmission of an earlier submission. The following is a list of the peer review reports and author responses from that submission.

Round 1

Reviewer 1 Report

Lee et al. submitted manuscript concerning the differential expression of several genes known to be correlated with colon cancer pathogenesis. Based on clinical material derived from patients-derived cancer samples representing various stages of CRC, the authors compared the expression of various genes such as APC, TP53, KRAS, PIK3CA, SMAD4, NRAS and others. It should be underlined that all of them were previously intensively studied and there are several papers describing their differential expression in context of tumor development. Thus, the novelty of this subject is quite limited.

According to Table 4 the 6 most common mutations is detected in the same genes: APC, TP53, KRAS, PIK3CA, SMAD4, FBXW7, regardless of tumor stage. The authors compared also data from biological samples with available databases. The significant difference refers in fact only to BRAF expression.

Presented data are only the results of genotyping analysis and confirm already known facts without any discussion that could confront them as an important factor for ex: chemotherapy response or other clinical outcome.

Minor comments: Supplementary data should also be formatted according to IJMS guidelines (Tables).

References do not follow IJMS guidelines too.

Reviewer 2 Report

Comments

The authors analyzed the genomic profiles of 95 pairs of primary and metastatic colorectal cancers.  Revealing the genetic profiles in metastatic CRC compared with primary CRC lesions was important data. The paper is well written; however, I have several questions that I would like to see addressed for further improvement of this manuscript.

Major Comments

There were several reports that analyzed the genomic alteration between primary and metastatic colorectal cancers. Although the authors described the differences from TGCA data, I would like to recommend clarifying the novelty of this study more clearly.

The driver genetic alterations involving colorectal carcinogenesis is an interesting finding, however, recently several studies reported that other factors such as methylation or tumor microenvironments also could affect cancer initiation and progression. When we decide on a treatment strategy, we have to take into account not only genetic mutations but also other factors. it may be easier to accept if the authors described this point in a more gentle tone especially in the conclusion.

It was confusing whether the point of this analysis was focused on the comparison between paired 95 primary and metastatic CRC or metastatic CRC including 47 metastatic only patients and 4 primary only CRC patients.

Minor Comments:

Unfortunately, I could not found the explanation of figure 4 in the main text. I was confused and wondering the explanation of figure 3 regarding TCGA data was actually figure 4. Generally, I would like to recommend including all figures in the main text.

In all 142 patients included this study, 95 patients have pair of primary and metastatic lesions. It is difficult to understand how the remaining 47 patients affected the results of this study. It would provide a better understanding that the authors annotated which data is a pair of 95 patients and which data is only metastatic data of the remaining 47 patients in the figures and tables.